# "...In Kiambu county, there are different things being done": A qualitative exploration of healthcare workers' experiences with cord care practices

**James Maina Githinji[1,2], Angeline Chepchirchir [3], Brian Ombura Nyanchoka[4], Prabhjot Kaur Juttla [5]\*, Ruth Nduati[2]**

1 Department of Health, County Government of Kiambu, Kiambu County, Kenya, 2 Department of Pediatrics and Child Health, Faculty of Health Sciences, University of Nairobi, Nairobi, Kenya, 3 Department of Nursing, Faculty of Health Sciences, University of Nairobi, Nairobi, Kenya, 4 Department of Pharmacy, Jomo Kenyatta University of Agriculture and Technology, Nairobi, Kenya, 5 School of Medicine, Faculty of Health Sciences, University of Nairobi, Nairobi, Kenya

\* pkjuttla13@students.uonbi.ac.ke

## Abstract

### Introduction

Neonatal sepsis (NNS) continues to be a leading cause of neonatal mortality in low-resource settings. NNS can occur secondary to umbilical stump infections (omphalitis) and this requires preventative cord care practices. To address this, the World Health Organization (WHO) issued context-specific recommendations for the use of 7.1% chlorhexidine (CHX) solution in areas with high neonatal mortality. Despite adoption of this policy, monitoring and evaluation of this guidance is lacking. This study describes the experiences of healthcare workers (HCWs) regarding cord care practices in Kiambu County, Kenya.

### Methods

We employed an exploratory qualitative design. Key informant interviews were conducted with 38 HCWs between October 6th and November 9th 2022 in six healthcare facilities in Kiambu County. Data were analyzed thematically using NVivo software.

### Results

While Ministry of Health guidelines endorse WHO's recommendation for the nation-wide use of CHX for neonatal cord care, its implementation in Kiambu county remains varied. HCWs continue to favour surgical spirit, and one of the healthcare facilities in our study adopted a facility-based policy of dry cord care. HCWs also relayed that mothers often combined modern and traditional methods. Some HCWs

**Data availability statement:** All relevant data are within the paper and its Supporting Information files.

**Funding:** This study was funded by NIH Fogarty Grant R 25TW011212 awarded to Prof. Ruth Nduati. The opinions and conclusions are by the authors and do not necessarily represent the funding institution position.

**Competing interests:** The authors have declared that no competing interests exist.

reported high satisfaction with CHX due to its perceived effectiveness, ease of use, and faster stump healing. However, systemic and product-specific barriers to uptake were noted. Systemic issues included frequent CHX stock-outs and high out-of-pocket costs due to inconsistent county-level supply and lack of insurance coverage. Product-specific concerns such as difficulty of application and reported adverse effects (e.g., delayed cord detachment, sepsis, burns) further impeded use. Knowledge gaps and contrary instructions among and between both the HCWs and mothers also undermined CHX use.

## Conclusion

Despite CHX being the recommended intervention for cord care in Kenya, its inconsistent use in Kiambu county reflects gaps in policy enforcement, supply chain reliability, and health education. The persistence of outdated or non-recommended cord care practices underscores a critical lack of oversight. To realize CHX's full public health potential, urgent policy action is needed to ensure consistent availability, enforcement of national neonatal guidelines, and investment into inclusive IEC materials. Bridging the gap between policy and practice is essential for reducing neonatal morbidity and achieving equitable progress towards Kenya's goal of Universal Health Coverage.

## Background

Neonatal sepsis (NNS) is a significant cause of neonatal mortality during the first month of life [1]. One of the primary contributors to NNS is the development of omphalitis, which is the infection of the umbilical stump [2]. This infection often arises as a result of improper and/or inadequate umbilical cord care practices, which can ultimately lead to neonatal mortality in up to 15% of cases [1]. While this is relatively uncommon in developed nations, it is a significant contributor to neonatal mortality in developing nations [1], such as Kenya.

Due to this burden of NNS secondary to omphalitis, the World Health Organization (WHO) has provided context-specific recommendations for newborns in settings where: (1) harmful traditional substances are prevalent and (2) the neonatal mortality rate (NMR) exceeds 30 in 1000 live births [3]. This recommendation is based on a meta-analysis of individual patient data, involving 177 neonates from five trials with a total of 129,391 newborns [3]. CHX for cord care has been shown to reduce NMR by up to 23% and prevents infection by 38–68% [4].

Furthermore, observational studies from low-income countries suggest that replacing harmful traditional cord care practices with antiseptics could save many newborn lives use [5,6]. However, this recommendation remains tentative due to the lack of data on sepsis and an inadequately described search strategy in the systemic review cited [5,6]. Moreover, the exclusion of observational studies from low-income settings (due to the absence of randomized controlled trials) limits its relevance for LMICs. The recommendation of dry cord care for LMICs that otherwise meet the criteria

is therefore questionable, especially since the Cochrane review itself found no significant difference in infection rates between placebo and antiseptic use [5,7].

Kenya has a 29.3% prevalence of NNS [8], and the incidence rate of omphalitis is estimated at between 20–77 of every 1000 live births of newborns delivered within healthcare facilities [9]. The incidence rate of omphalitis is purported to be higher in at-home births [9]. The NMR has been documented 21 deaths per 1,000 live births, accounting for 66% of infant deaths and 51% of under-5 deaths [10]. However, this NMR varies widely between the counties of Kenya. A 2018 study conducted in Machakos County revealed that, of all the admissions in the newborn unit, 34% constituted cases of NNS and were associated with 15% of the deaths reported [9].

WHO's recommendation for the contextual use of CHX for cord care is a transition from the previous recommendation of the use of surgical spirit [11], a recommendation which was also reflected in Kenya's 2009 National Guidelines for Quality Perinatal Care [12]. Surgical spirit is no longer recommended for umbilical cord care due to its limited efficacy in preventing infections, potential to irritate newborn skin, and the availability of safer, more effective alternatives like CHX [11,13–15]. In 2016, however, the Government of Kenya recommended the use of CHX in all births to ensure equal standards in health-care provision for all citizens of the country [16], therefore adopting the new WHO guidelines nation-wide [17].

Since Kenya's adoption of the WHO recommendation on the use of CHX in all neonates in 2016, the implementation has been set to take three phases: (1) preparation, (2) planning, and (3) execution. Expounded, this process includes the alignment of policy, developing guidelines, introducing the product, training of service providers, generating demand for the product, and finally, monitoring and evaluation (M&E).

A 2016 study by Muriuki *et al.* found that HCWs in rural Kenya widely accepted the use of CHX for umbilical cord care, citing benefits such as faster healing, reduced infections, ease of use, and positive community feedback [18]. For successful scale-up, they emphasized the need for community sensitization, proper training, consistent supply, clear guidelines, and inclusive service delivery approaches [18]. However, a 2021 cross-sectional study at Kangundo Level 4 Hospital found that while most mothers received some education on CHX use, both mothers and HCWs demonstrated poor understanding of its proper application [9].

Different counties are expected to have achieved different levels of adoption given that healthcare in Kenya is devolved to the county level with the national level providing policy guidance. Given that Kenya has a goal to further increase the uptake of CHX treatment to above 80% in all health care facilities by 2026 [4], the adoption of CHX for cord care has yet to undergo M&E through the perspectives of HCWs. The CHX uptake in Kiambu county is 40.1% [18], and therefore, understanding HCW experiences with cord care practices in the county would be essential to shape health policy and highlighting gaps in its implementation.

## Methods

### Study design

The study adopted an exploratory qualitative design using key informant interviews (KIIs) guided by semi-structured question guides, and which were conducted between 6th October 2022–9th November 2022. The data was collected from six health facilities in Kiambu County.

### Study setting

This study was carried out in Kiambu County, Kenya, which has a population of over 2.4 million people. The location coordinates are 1⁰ 10'S 36⁰50'E, and it neighbors six other counties, namely Nairobi City and Kajiado Counties to the South, Machakos to the East, Murang'a to the North East, Nyandarua to the North west, and Nakuru County to the West [19]. The county is 60% urban and 40% rural due to its proximity to the capital city Nairobi.

The county has 505 health facilities, comprising 108 public, 64 faith-based, and 333 privately owned facilities. In Kenya's health system, Level 1 refers to community-based services (e.g., health promotion and preventive care), Level 2 to dispensaries, Level 3 to health centers, Level 4 to sub-county hospitals, and Level 5 to county referral hospitals.

Maternity services are typically offered at Level 3 (health centers) and above, including private and faith-based hospitals. Therefore, this study gathered insights from HCWs working in Level 3, 4, and 5 public facilities, as well as private hospitals. One of the Level 5 hospitals (Thika Level 5 Hospital) in this study also functions as a teaching hospital for medical and nursing students.

## Local context

CHX for newborn umbilical cord care is available in two formulations: gel and solution [20]. Both forms are recommended by the WHO and are included in the Kenya Essential Medicines List as well as in postnatal care guidelines [20–22]. In line with this, Kenya's Ministry of Health (MoH) endorses the availability and use of both CHX formulations [20].

The choice between the dosage formulations used depends on what is acceptable to mothers, caregivers, HCWs and availability [23].

According to the national guidelines, immediately post-delivery, a HCW should apply CHX to the newborn's umbilical cord using sterile gloves. The antiseptic is applied to the base, stump, and tip of the cord [20]. For gel formulations, the HCW uses their index finger to spread it; for the solution, the dropper is used to apply CHX directly. The CHX is not cleaned off after application. In the postnatal period, CHX should be applied once daily until the seventh day or until the cord naturally detaches: whichever happens first. Mothers and/or caregivers are instructed on how to apply CHX at home following the same procedures, but *in lieu* of sterile gloves, they are advised to thoroughly wash their hands before each application.

The National Health Insurance Fund (NHIF), now defunct, but which was active during the period of the study, had a program called *Linda Mama* that covered the costs of antenatal care (ANC), delivery, and postnatal care (PNC) for all pregnant women [24]. However, it notably excluded the cost of CHX, and currently, there is no government-sponsored program providing CHX free of charge to mothers during the postnatal period.

## Study population and sampling

The study population consisted of over 2,600 HCWs providing health services in Kiambu County during the study period. No formal sample size was calculated and we applied the information saturation principle [25]. Two providers were targeted in every unit (with a target of 30 HCWs) during the study's formulation, however some units were understaffed during the study and hence only had one HCW interviewed. The cadre and respective facility of the sample are presented in Table 1.

*Inclusion criteria* - HCWs who had direct interactions with neonates or were involved in providing or handling CHX cord antiseptic during the study period, and who had at least 6 months of experience with cord care.

**Table 1. Facility sampled and cadre of the healthcare workers interviewed who are involved in the continuum of cord care.**

| Facility type | Name of facility | ANC/PNC nurse | Midwife | Pediatrician | Pharmacist |
|---|---|---|---|---|---|
| Level 3 | | | | | |
| | Githunguri | 2 | 2 | 1 | 1 |
| Level 4 | | | | | |
| | Kihara | 2 | 1 | 1 | 2 |
| | Ruiru | 2 | 1 | 1 | 1 |
| Level 5 | | | | | |
| | Thika | 2 | 1 | 3 | 1 |
| | Kiambu | 1 | 3 | 2 | 2 |
| Private | | | | | |
| | Nazareth | 1 | 2 | 2 | 1 |

 

*Exclusion criteria* - HCWs who were not directly involved in newborn care or who declined to provide informed consent and who had less than 6 months of experience with cord care.

## Data collection

A total of 38 KIIs were conducted between 6th October 2022 and 9th November 2022.

All interviews were conducted by two members of the study team, JMG and BNO, both holding Bachelor of Pharmacy (BPharm) degrees. JMG serves as the Kiambu county pharmacist and holds a PhD in pharmacy, while BNO is a researcher with experience in qualitative data analysis. Both researchers received training prior to data collection to ensure consistency and quality.

The interviews were conducted in English, but participants were allowed to respond in either English or Kiswahili. BNO, who is bilingual and fluent in both written and spoken English and Standard Kiswahili, translated the KIIs, which were audio-recorded and transcribed verbatim directly from Kiswahili into English when applicable.

Each participant was assigned a unique index number (1–38), and quotations are referenced using the format: HospitalName_Cadre_IndexNumber.

## Data management

Kobo Toolbox software was used to collect and manage data, with all data coded to conceal the participants' identities. To ensure data security, electronic data collected via KoboCollect were encrypted and password-protected on the devices. After transfer, data files were securely stored on password-protected computers and backed up on a secure cloud platform with restricted access. NVivo project files were similarly protected to maintain confidentiality and data integrity throughout the analysis process.

## Data analysis

Interview transcripts were imported into NVivo software to assist in the classification, sorting, and coding of data into central themes and subthemes. The coding process followed a two-cycle approach. In the first cycle, value coding was used to capture participants' attitudes and values, reflecting their perceptions and experiences. The second cycle involved focused and pattern coding. Focused coding developed specific categories without focusing on their properties, while pattern coding grouped similar codes into meaningful units for theme identification [26–28].

NVivo's platform played a crucial role in facilitating this multi-level coding process, ensuring consistency in the analysis framework by identifying recurring patterns and connections. The research questions and interview guide informed the coding frame, allowing for accurate classification of notable data segments to reveal dominant themes. The integration of value, focused, and pattern coding in both cycles ensured a comprehensive and structured analysis of the interview data [26–28].BNO, JMG, and PKJ also independently reviewed the transcripts to familiarize themselves with the data and identify initial codes. Discrepancies were resolved through discussion to ensure consistency and reliability.

## Ethical considerations and approvals

Upon arrival at the various study sites, verbal consent was initially sought from the manager of each facility unit, followed by written consent from the hospital medical superintendent.

Prior to the initiation of data collection, participants were informed about the study's objectives and their role as respondents. They were assured that participation was entirely voluntary and that they had the right to withdraw at any time. All data collected by the researchers were treated as confidential and utilized exclusively for this study. Access to the materials was restricted to the investigators, and stringent data management protocols were implemented to uphold the confidentiality of the study subjects.

Approval was obtained at various administrative levels, including county and sub-county authorities before the study commenced. Ethical clearance was secured from the University of Nairobi/Kenyatta National Hospital Ethical Review Committee [P470/05/2022], County Government of Kiambu Department of Health Services, Kiambu County [KIAMBU/HRDU/22/09/14/RA_GITHINJI], and a research license was obtained from the National Commission for Science, Technology and Innovation [NACOSTI/P/22/20446].

## Results

### Demographic characteristics of participants

A total of 38 HCWs were interviewed, as depicted in Table 1. The mean age of respondents was 35.6 years (Range: 23–57 years). The HCWs interviewed had a mean of 9.4 years of professional experience, with a median of 7.5 years, a mode of 7 years, and a range spanning from 0.5 to 30 years. In terms of time spent at their current stations, the average was 4.4 years, with a median of 3 years, a mode of 1 year, and a range from the minimum of 0.5 years to 22 years. Participants had an average practice duration of 9 years.

### Findings

The transcripts drawn from recordings of KIIs held with selected HCWs were analyzed, and themes emerged as 33 files with 52 references. Some themes mirrored the guiding questions in the semi-structured questionnaire, as shown in the coding tree as Fig 1.

### A. Cord care practices in Kiambu county

#### 1. National policy recommendation

We sought to understand the general cord care practices in various health facilities across Kiambu County by interviewing HCWs. According to MoH guidelines, the use of CHX for cord care has been officially recommended, with the directive being disseminated from the central government to the county level. This was confirmed by a participant from Nazareth Hospital, who stated, "*first of all, we have the policy that is an order that came from Kiambu County from the Ministry of Health that chlorhexidine should be used*" (**Nazareth_Pharmacist_25**).

The majority of participants acknowledged the existence of the MoH guidelines recommending CHX for cord care, confirmed that this policy had been communicated to health facilities and gave a description of the same policy in their responses.

*"So after birth, when you cut the cord, immediately we use the chlorhexidine, and after, we use it once a day, once a day after delivery until now the mothers goes home and you instruct them to use it until the cord dries off and falls off."* **(Nazareth_Midwife_27)**

*"…the next thing is to maintain hygiene of the cord, and this is done by applying an antiseptic. Commonly we use chlorhexidine, either the gel or the solution." (***Kiambu_ Paediatrician_36)**

*"...The guys from the* [MoH] *when they come, they say, "No, you must use [CHX]. You must use [CHX]..."* **(Thika_Midwife_13)**

#### 2. Facility-level departures from national recommendations

HCWs mentioned that surgical spirit remained the predominant choice in practice, indicating a reluctance to fully transition from previous guidelines. In fact, almost half (41.7%) of participants reported having prior experience with surgical spirit before being required to switch to CHX, which was a new product in their facilities. Some facilities reported not using CHX at all, relying solely on surgical spirit for cord care. This variation in practice highlights the need for further investigation into the slow adoption of CHX in these settings, despite the existence of formal guidelines.

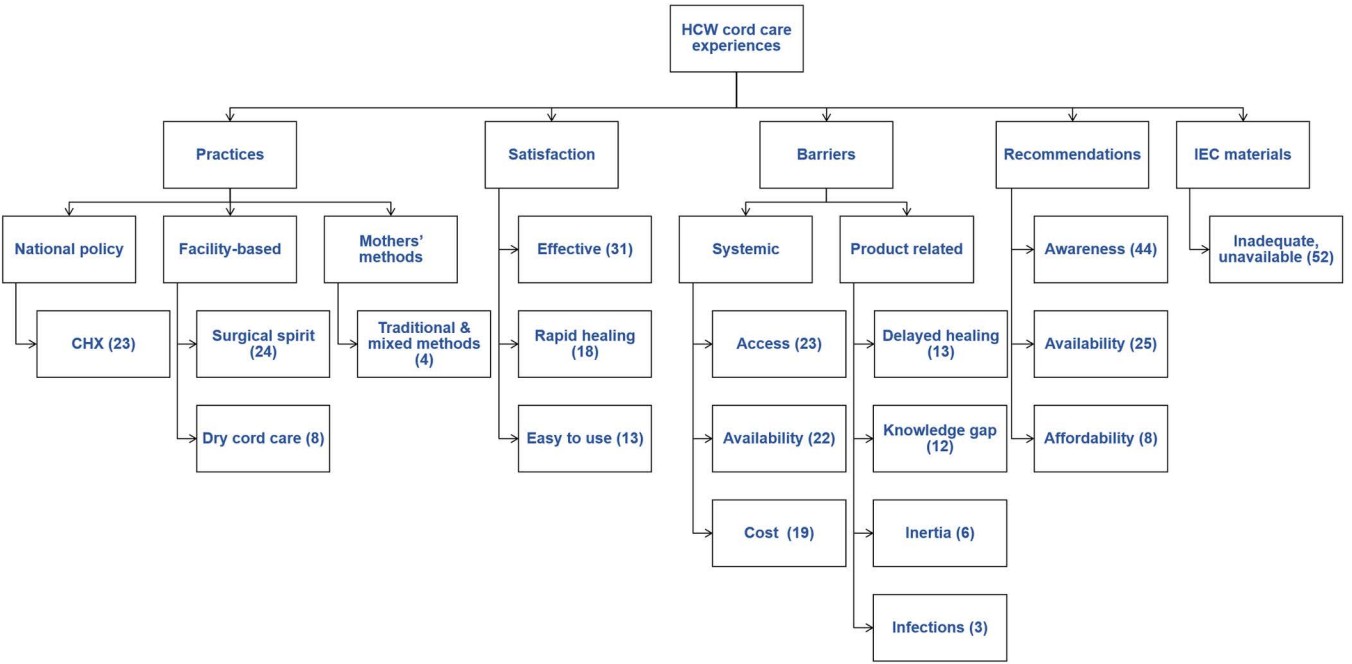

**Fig 1. The coding tree of transcripts from interview audio recordings.** The numbers in brackets indicates the total number of individual references that mention that specific code within the files.

Furthermore, despite clear directives, Thika Level 5 Hospital actively deviated from the National Guidelines, opting instead to follow its own facility-based protocol of dry cord care.

*"Here in our health facility... We discussed with our pediatrician and we just decided to leave the cord just like that. Don't put anything, don't put water, don't put the spirit, and don't put chlorhexidine, as long as you keep it clean. And we have not had any case of sepsis... The guys from the* [MoH] *when they come, they say, "No, you must use* [CHX]. *You must use* [CHX]*." But then we are agreed for us, we don't use it here, we have not been using it, and we have not had any cases of neonatal sepsis."* **(Thika_Midwife_13)**

*"So the current cord care practice in Thika Level 5 I would say, predominantly because that's where I practice is dry cord care. I hardly do surgical spirit or chlorhexidine or any other medication directly with the cord."* **(Thika_Paediatrician_12)**

*"And what I've noted, there is less uptake of chlorhexidine use for cord care compared to the surgical spirit.... the pediatrician and also other practitioners [are] persistent on the use of surgical spirit as opposed to chlorhexidine."* **(Nazareth_ANC-nurse_23)**

*"So currently, in Kiambu County, there are different things that are being done … some places and institutions that are using chlorhexidine for cord care. Then a number of institutions are using surgical spirit. There are some people who even believe on doing nothing to cord care. There are people who are using normal saline for cord care."* **(Thika_Paediatrician_37)**

*"…I just know most of the time, they don't even use that chlorhexidine. They use surgical spirit."* **(Kihara_Pharmacist_18)**

### 3. Mothers' use of blended and traditional cord care methods

A minor theme emerged in which some HCWs reported that mothers used multiple cord care methods, often influenced by either their past experiences or family advice, or both. For example, if surgical spirit had been ineffective with their first baby, they would switch to CHX for subsequent children, and vice versa. In some cases, mothers who delivered during the era of previous guidelines recommending surgical spirit continued to favour it. Other practices included the use of saliva, a traditional method still common in our context.

> "*There is the use of … saliva, especially for the traditional mothers*" **(Ruiru_PNC-nurse_5)**

> "*One challenge is that mothers out there, they have other methods of doing cord care. When you teach them how to do it, they go at home and they're taught by other relatives how to do it … Like they will buy spirit because the other baby they used spirit, or they will use another thing like savlon that's hindering proper use of the chlorhexidine.*" **(Nazareth_ANC-nurse_23)**

> "*some mothers are resistant to change, especially those who delivered much earlier and chlorhexidine was not there before. So if they used spirit, they still want to use spirit … or even using saliva or cow dung.*" **(Nazareth_Midwife_27)**

> "*The common challenge that we face is that mothers believe in the use of some traditional methods like use of saliva.*" **(Ruiru_Midwife_6)**

### B. HCW satisfaction with CHX for cord care

The preferences that HCWs had for CHX in cord care were varied, largely shaped by their clinical and anecdotal experiences with different methods. Approximately half of the HCWs who expressed satisfaction with CHX did so primarily due to its perceived effectiveness in preventing omphalitis and NNS. This ultimately reduced the need for additional healthcare visits for both mothers and newborns.

Many HCWs in favor of CHX also highlighted that it promoted quicker stump healing compared to other methods, provided that mothers followed proper instructions, maintained good hygiene, and began cord care immediately or shortly after birth. Additionally, CHX was praised for being easy to use, contributing to its widespread adoption and positive outcomes.

> "*So far, the outcomes have been good. We have had limited cases of sepsis after the use of chlorhexidine.*" **(Kiambu_Midwife_22)**

> "*… those who practice, okay, cord care with chlorhexidine as they are instructed, the outcome is good.*" **(Kihara_Midwife_16)**

> "*The use of chlorhexidine for cord care has made work easier in Kiambu since it's easy to use and its availability, its efficacy being high has made it easier for us to manage the newborns and their mothers are even happy with the healing process using chlorhexidine gel.*" **(Kiambu_Pharmacist_34)**

> "*The effectiveness of cord care is better compared with [traditional or surgical spirit] methods … in comparison with the mothers that have been using the surgical spirit and the cord care, during the postnatal checkup one week post-delivery, the mothers who have been using the chlorhexidine, the cords are healed faster compared with other methods.*" **(Ruiru_PNC-nurse_5)**

### C. Barriers to CHX adoption and effective use for cord care

#### 1. Systemic barriers

One of the most significant barriers to the adoption and provision of CHX for cord care was its inconsistent availability in government health facilities. This challenge of access was twofold: not only was there a shortage of

government-supplied CHX to health facilities, and when it was available, its cost often priced consumers out of the market. Many mothers at the facility found the cost of CHX prohibitive. As a result, HCWs recommended more affordable alternatives, which hindered CHX's widespread adoption and led mothers to choose cheaper options like surgical spirit.

> "…And then the erratic supplies for the county, particularly because of those financial constraints the counties have been having have made it easy for us to try out other things because in the absence of chlorhexidine, what else was being done to allow cords to heal... the supplies and the county, it's quite irregular. The stock-outs are very, very, very common because of the budgetary constraints in the county." **(Thika_ Paediatrician _12)**

> "The first challenge is limited resources from the county. That is, when receiving our delivery from KEMSA, we might also be able to receive chlorhexidine. Another challenge is from the patients. Once the drug is prescribed, some patients may not be able to purchase from the pharmacies outside." **(Kihara_Pharmacist_24)**

> "The challenges of provision of chlorhexidine gel, first of all, I'd say it's not readily available. As in, even if I sent a mother outside here to buy, I don't think it's as available as there rest. Secondly, the cost. It's a challenge because the mothers are not able to afford it" **(Githunguri_ANC-nurse_33)**

## 2. Perceived product-specific disadvantages

Some HCWs associated CHX use with delayed cord healing, sepsis, and prolonged cord detachment. In a few instances, they reported that neonates had returned to the hospital with infections after its use. Concerns also emerged over the varying CHX formulations, which caused confusion, particularly due to its resemblance to tetracycline eye ointment and challenges in proper application. The drop formulation was especially scrutinized, with some HCWs believing it contributed to sepsis and soft tissue burns. Their perceptions of CHX's effectiveness were largely shaped by mothers' feedback. These second-hand accounts reinforced the belief that CHX was less effective than other cord care methods, ultimately influencing its use within health facilities.

> "…The [CHX] drops, the mothers are coming back with septic cords. I don't know whether it's the hygiene at home or what. I don't get. But when we get a mother that comes and then we take the history, they say they bought the drops. But with the gel, no, not much sepsis. But with the drops, there is." **(Kiambu_Midwife_35)**

> "Now, the problem with the gel was that either it was too thick and it kept leaving the cord moist, and so it would take longer for the cord to drop off … So, a lot of children would come with delayed cord dropping." **(Thika_Paediatrician_12)**

> "Initially, we used to use the chlorhexidine gel, which was a bit difficult to use because it required one to wash their hands or change their gloves, apply the gel on their finger, and then spread the gel on the cord and surrounding skin. The gel could also be confused with tetracycline eye ointment because of the kind of packaging it came in. However, with the solution, it's much easier to use because all one needs to do is to drop a few drops on the umbilical cord and leave the baby's cord to dry. The packaging is also very different from the tetracycline eye ointment, so confusion is reduced." **(Kiambu_Paediatrician_36)**

## 3. Inconsistent use instructions

The data corpus revealed a significant knowledge gap among HCWs and mothers regarding the use of CHX for cord care, with some indicating that it should be applied once a day but others recommending it twice a day. HCWs often demonstrated preferences for alternative practices, reflecting a reluctance towards the adoption of CHX. These gaps were primarily due to insufficient HCW training on CHX application, leading to a lack of awareness among mothers about proper usage.

"*… the mothers need an education. There are mothers who will not want to use chlorhexidine gel because of mythical thoughts or superstition, or that's not what they were taught by their mothers to use. So, sensitization is needed … we need to sensitize the healthcare workers because sometimes we have found mothers who have not used chlorhexidine gel because the physician who was working with the mother did not believe in the working of chlorhexidine gel.*" **(Kiambu_Pharmacist_34)**

### 4. Inertia

Among HCWs, the knowledge gap regarding CHX use is further complicated by resistance to switch from surgical spirit, which many have successfully used in the past. This resistance is often rooted in personal experiences with both products. HCWs tend to favor surgical spirit, largely due to their familiarity with it and the perceived shortcomings of CHX.

"*I think the fact that most of the healthcare workers were used to surgical spirit, the switch to chlorhexidine and the fact that it takes time, the cord takes time to heal, I think most people are still reluctant to start mothers on the gel. They'll prefer surgical spirit.*" **(Kihara_PNC-nurse_15)**

### D. Recommendations to increase CHX uptake

HCWs underscored the need for increased awareness and education among both HCWs and patients, citing persistent knowledge gaps regarding CHX's benefits and proper usage. Many recommended improving accessibility by including CHX in maternity packages or ensuring its consistent availability in hospital pharmacies, thereby enabling timely application immediately after delivery, which is a critical factor for its efficacy.

Affordability also emerged as a central concern, with suggestions to reduce the cost of CHX and incorporate it into health insurance schemes such as (formerly) *Linda Mama* to alleviate the financial burden on mothers. Furthermore, HCWs advocated for comprehensive training programs targeting midwives, doctors, and pediatricians to deepen their understanding of CHX. Such training would enable healthcare providers to more effectively counsel patients, thereby promoting greater uptake of CHX in cord care practices.

"*Yes. So number one, it's the cost. It is readily available to them considering that it is covered by NHIF, number one...*" **(Nazareth_Midwife_27)**

"*Suggest that the hospital should source for more of the drug to ensure that it's constantly available for use to increase its uptake.*

*Interviewer: If it was available, do you think everybody would use.*

*Interviewee: Yeah.*" **(Kiambu_Midwife_22)**

"*I think we need proper training for the staff, especially in maternity, even for us working in the postnatal clinics, because I understand the gel should be started immediately after birth.*" **(Kihara_PNC-nurse_15)**

"*The other one is educating, first of all, the healthcare worker on how to use it so that the healthcare worker is empowered to also educate the mothers on how to use it.*" **(Thika_Pharmacist_11)**

### E. Opinions on information, education, and communication (IEC) materials

IEC materials depicting the application of CHX were found to be largely unavailable and inadequate in healthcare facilities. A significant majority of respondents (86.1%) reported that these materials were inaccessible in most hospital units, resulting in ineffective promotion of CHX utilization. In cases where materials did exist, they were often not suitable for illiterate women, who frequently resorted to traditional methods of cord care.

To address this gap, HCWs suggested developing more inclusive IEC materials, such as pictorials and patient-friendly infographics, to ensure that all mothers could benefit from the health education provided. These materials included charts, pictograms, and illustrations that use inclusive language. using accessible language. These patient-friendly IEC materials should be strategically placed in locations such as antenatal clinics and labor wards, so that mothers are informed about the importance of CHX and can prepare to obtain it before delivery. HCWs also recommended tailoring resources specifically for their needs, including physical charts as reminders at workstations, along with Continuous Medical Education (CME) sessions and training programs. Observations made by the participants included the following:

*"But in the facility in Githunguri, I've not seen any [IECs]. And I think CME is about the chlorhexidine would really help in creating awareness… everywhere, in every ward, in the postnatal rooms, and in labor ward, and also here in antenatal."* **(Githunguri_PNC-nurse_29)**

*"On the issue of IEC materials, they're not available in our facility. If they can be effective and brought down to the facilities, I would recommend they be more pictorial. By a glance, someone is able to know what it is and how it can be used and the readings big enough to be legible."* **(Githunguri_ANC-nurse_33)**

*"The IEC material would help the staff here be at least more conversant for those who are not conversant with chlorhexidine gel, its use, its application, the period which the patient is supposed to use... Through the information the staff acquire, they'll be able to make more informed decisions of even escalating the matter to facilitate the increase of the supply of the chlorhexidine gel in the facility and also acts a refresher …as opposed to other previous methods that were used of cord care…The IECs are not available at the facility at the moment..."* **(Kiambu_Midwife_22)**

*"For the charts, they should be bigger, and they should provide more charts so that we can put maybe in the wards and to educate the mothers. And for those mothers who are illiterate who cannot read, the wordings are many than the pictures. The pictures are better illustrating of how to use the chlorhexidine."* **(Nazareth_Midwife_27)**

## Discussion

This study explored the experiences of HCWs in Kiambu County regarding newborn cord care and revealed significant deviations from Kenya's national guidelines which recommends the blanket use of CHX for all neonates nationwide. Despite the increasing adoption of CHX, with 41.7% of HCWs reported transitioning from surgical spirit, outdated and potentially harmful practices such as the use of surgical or methylated spirit persisted in our set-up. One facility in our study, Thika Level 5 Hospital, adopted its own facility-based practice of dry cord care. Our findings suggests that the national policy on CHX has not been fully implemented or embraced in practice, underscoring the need for strengthened policy adherence and education.

The participants in this study reported diverse experiences with the use of CHX for cord care. Key concerns included the cost of CHX products, delayed healing, and varying perceptions of its effectiveness. The cost of CHX, which was quoted between Ksh. 120–200 (USD 0.93 to 1.55), was perceived as expensive by some clients, particularly when compared to more affordable alternatives such as methylated spirit (Ksh. 50, USD 0.34) or dry cord care, which incurs no cost. This contrasts with the findings of Opanga et al., who reported that CHX gel was available at less than Ksh 40 in private facilities and pharmacies, and was free in public hospitals, yet with only a minimal uptake of 1.3% [29]. Another study conducted in rural Bungoma county found that CHX was provided free of charge to mothers after delivery, but noted that improper use and misunderstandings regarding its application negatively impacted uptake, along with echoing concerns about delayed cord separation [30].

The variations in participants' experiences with CHX use in Kiambu County may reflect the unique socio-economic and health system dynamics of a peri-urban area situated near the capital city. While Kiambu's proximity to Nairobi may

suggest better access to health information and services, it also presents a complex interplay between urban and rural health-seeking behaviours. Unlike rural Bungoma, where CHX was provided free and uptake issues were mainly due to poor understanding or instructions, Kiambu residents may encounter CHX more frequently through private pharmacies or clinics where it is sold at significantly higher prices, making it prohibitive to lower-income households despite theoretical availability. Furthermore, price discrepancies between public and private sectors may indicate a lack of standardized pricing or public awareness about where to obtain CHX for free. This is supported by the findings of Opanga et al., which noted low uptake despite free availability in public hospitals [29]. This demonstrates a disconnect between policy and implementation, possibly fueled by inadequate dissemination of information, lack of health worker engagement, or supply chain challenges in public facilities.

Despite these challenges, our results also highlighted positive feedback from HCWs, particularly around the ease of use, prevention of sepsis, and faster cord separation, suggests potential benefits. This echoes a participant in Opanga et al.'s study who reported early cord separation, with the cord drying and detaching after just three days, highlighting the promise of CHX in improving neonatal outcomes when used appropriately [29].

The majority of HCWs interviewed indicated challenges in the provision of CHX due to systemic challenges of unavailability. The inclusion of CHX in the Kenya Essential Medicines List was intended to enhance its availability in the public sector, and this highlights a gap in enforcement of this directive in our context [22]. The overarching goal of healthcare systems is to ensure that all patients have consistent access to a comprehensive range of suitable, high-quality, and affordable essential medicines, as emphasized by the WHO [31]. Essential medicines are critical components of healthcare, recognized alongside clean water, safe food supply, and adequate housing as fundamental necessities; their absence constitutes a failure to uphold universal human rights [32,31].

However, essential medicines frequently experience stock-outs in developing countries [32,31]. In Kenya, where health services are devolved (the national government "shares" government functions with smaller county governments), the impact of these stock-outs varies across the different counties. To mitigate the occurrence of stock-outs for essential medications on the package list, it is crucial for public sector facilities to ensure timely payments to the Kenya Medical Supplies Authority (KEMSA) for their orders [33,34]. Conversely, non-public facilities report infrequent stock-outs due to differing and strengthened procurement processes. Stock-outs were also identified as a significant concern by Ambale *et al.* at Kangundo Hospital [16]. Consequently, participants reported a lack of sustained experience with the products, which undermined their confidence in persuading mothers to purchase them upon leaving the healthcare facilities. Furthermore, to achieve optimal coverage, particularly for economically disadvantaged families, subsidization may be necessary [35].

The aforementioned issue raises a critical discussion about the disconnect between policy and implementation under Kenya's devolved health system. It highlights that simply listing a drug as essential is not enough without effective supply chain management, accountability structures, and financing mechanisms at the county level. The situation in Kiambu highlights how inconsistent availability of essential medicines like CHX undermines provider confidence, policy adherence, and public trust, all of which contribute to missed opportunities for preventing neonatal infections and deaths. Moreover, the inequity between public and non-public facilities, where the latter have more consistent procurement systems, underscores the urgency of reforming public sector logistics and financing flows: particularly timely disbursement of funds to suppliers. Without these structural changes, the promise of universal health coverage and improved neonatal outcomes will remain unrealized. This discussion also sets the stage for recommending policy enforcement mechanisms, decentralized budgeting reforms, and stronger accountability for county health departments, as essential steps toward resolving persistent stock-outs of life-saving commodities.

HCWs reported inadequate training in CHX cord care, leading to significant knowledge gaps and unfamiliarity with recommended guidelines. This resulted in variations in the reported frequency of CHX application. This finding aligns with another Kenyan study, which linked the lack of training and clear instructions to the ineffectiveness of CHX gel [16]. To address research-driven shifts, standardized guidelines for CHX use are essential. Ongoing medical education for all

 

cord care personnel is crucial. Participants also emphasized the need for increased media publicity to enhance consumer knowledge, in line with the sensitization efforts recommended by Opanga et al. [29].

According to the Kenya Demographic and Health Survey (KDHS), 89.2% of births in Kiambu County occurred in hospitals, with 98% of these deliveries attended by medical professionals [36]. This rate is notably higher than the 54% and 64% reported in Pemba (Tanzania) and Zambia, respectively [37,38]. However, having a medically assisted delivery does not necessarily guarantee a reduced risk of infection, as many medical centers in our context fail to meet minimal infection prevention standards [39]. Furthermore, potentially hazardous procedures that raise concerns in home delivery settings are often performed on women and infants before they are discharged within hours of birth [15].

While Kiambu's high institutional delivery rate might suggest optimal conditions for newborn cord care, this can create a false sense of security if not paired with stringent infection prevention and control (IPC) measures. It challenges the assumption that health facility births inherently reduce neonatal infections and points to the critical role of CHX as a safeguard, even within supposedly safer environments. This further reinforces the WHO's position on universal CHX application, emphasizing that the presence of a skilled birth attendant does not automatically equate to safe neonatal outcomes: especially in under-resourced or overstretched facilities. Integrating CHX as a routine part of postnatal care, regardless of delivery setting, can act as a simple, cost-effective buffer against systemic IPC failures, especially in high-volume facilities that discharge mothers and newborns within hours of delivery.

Most HCWs in our study suggested several strategies to enhance the uptake of CHX. IEC is crucial for ensuring correct usage, infection control, patient safety, and adherence to best practices. Participants emphasized the importance of recommending CHX to mothers and providing education on its proper use. These sentiments are echoed in the existing literature as this resistance to CHX has been previously described in similar settings [13,30,40–42]. Both the quantity and quality of IEC materials on CHX need improvement. Visual aids and illustrations should be prioritized over text to enhance mothers' understanding of proper CHX use. Our study found that, while most health facilities had some IEC materials on CHX, these resources were limited and lacked engaging illustrations to support learning. This finding aligns with those of the Bungoma County study [30], and further highlights a pressing need to improve these materials for HCWs. While participants rightly identified the aforementioned methods as key to increasing CHX uptake, these strategies cannot succeed without a robust and context-sensitive health communication framework. The limited, often text-heavy IEC materials reported in this study reveal a missed opportunity to engage mothers, many of whom may have low health literacy or limited formal education, in ways that resonate with their daily realities. This challenge calls for a shift from generic IEC content to co-designed, visual, and language-appropriate tools that accommodate all literacy levels.

Devolution in Kenya offers both a challenge and an opportunity: counties like Kiambu can tailor CHX strategies to their unique population needs, but this must be done in alignment with national guidelines and messaging, to ensure consistency and avoid confusion. This requires capacity-building for local health workers, investment in community engagement strategies, and partnerships with trusted media outlets to amplify accurate, culturally resonant messages on CHX use. Ultimately, the effectiveness of CHX as a public health intervention depends not just on availability, but on how well it is communicated, understood, and trusted at the community level.

## Conclusion

The mixed perceptions, inconsistent use of CHX for cord care and the use of alternative cord care methods in Kiambu County reveal a deeper systemic issue that extends beyond individual preferences: it points to critical gaps in policy enforcement, supply chain reliability, and health communication. The fact that one of the facilities included in our study practice cord care contrary to national guidelines is particularly concerning and highlights a lack of oversight and accountability in guideline implementation. To ensure CHX fulfills its potential as a low-cost, high-impact intervention for preventing neonatal infections, there is an urgent need for policy action at both county and national levels. This includes: (1) strengthening the enforcement of national neonatal care protocols across all facilities, (2) ensuring uninterrupted

availability of CHX through timely procurement and subsidization, and (3) developing and disseminating culturally appropriate, visually driven IEC materials to improve community understanding and uptake.

As Kenya advances toward Universal Health Coverage, it is imperative that essential medicines like CHX are not only listed on paper but made reliably available and appropriately used at the point of care. A unified, well-supported CHX strategy will not only align with WHO recommendations but also directly contribute to reducing neonatal morbidity and mortality: especially among the most vulnerable populations. This must begin with aligning policy intent with health system practice.

## Supporting information

**S1 File. Raw data.** This document contains the transcriptions of the IDIs and the indices used for each participant. (PDF)

## Author contributions

**Conceptualization:** James Maina Githinji, Angeline Chepchirchir, Ruth Nduati.

**Data curation:** Brian Ombura Nyanchoka, Prabhjot Kaur Juttla.

**Formal analysis:** James Maina Githinji, Brian Ombura Nyanchoka, Prabhjot Kaur Juttla.

**Funding acquisition:** Ruth Nduati.

**Investigation:** James Maina Githinji, Brian Ombura Nyanchoka.

**Methodology:** James Maina Githinji, Brian Ombura Nyanchoka, Ruth Nduati.

**Project administration:** James Maina Githinji, Angeline Chepchirchir, Ruth Nduati.

**Resources:** Angeline Chepchirchir, Ruth Nduati.

**Software:** Brian Ombura Nyanchoka.

**Supervision:** Angeline Chepchirchir, Ruth Nduati.

**Validation:** James Maina Githinji, Angeline Chepchirchir, Prabhjot Kaur Juttla.

**Writing – original draft:** James Maina Githinji, Brian Ombura Nyanchoka, Prabhjot Kaur Juttla.

**Writing – review & editing:** James Maina Githinji, Angeline Chepchirchir, Brian Ombura Nyanchoka, Prabhjot Kaur Juttla, Ruth Nduati.

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
