## [Decision Letter · Decision Letter 0]

PONE-D-25-16839“...in Kiambu county, there are different things being done”: A Qualitative Study Exploring the Experiences of Healthcare Workers with 7.1% Chlorhexidine digluconate Cord Care in Kiambu County, Kenya

PLOS ONE

Dear Dr. Juttla,

Thank you for submitting your manuscript to PLOS ONE. After careful consideration, we feel that it has merit but does not fully meet PLOS ONE’s publication criteria as it currently stands. Therefore, we invite you to submit a revised version of the manuscript that addresses the points raised during the review process.

We look forward to receiving your revised manuscript.

Kind regards,

Sojib Zaman, MBBS, MSc., PhD

Academic Editor

PLOS ONE

 [This study was funded by NIH Fogarty Grant R 25TW011212 awarded to Prof. Ruth Nduati. The opinions and conclusions are by the authors and do not necessarily represent the funding institution position.]. 

Additional Editor Comments (if provided):

Reviewers' comments:

Reviewer's Responses to Questions

**Comments to the Author**

1. Is the manuscript technically sound, and do the data support the conclusions?

Reviewer #1: Yes

Reviewer #2: Yes

Reviewer #3: Partly

2. Has the statistical analysis been performed appropriately and rigorously? 

Reviewer #1: N/A

Reviewer #2: Yes

Reviewer #3: Yes

3. Have the authors made all data underlying the findings in their manuscript fully available?

Reviewer #1: Yes

Reviewer #2: Yes

Reviewer #3: Yes

4. Is the manuscript presented in an intelligible fashion and written in standard English?

Reviewer #1: No

Reviewer #2: Yes

Reviewer #3: Yes

5. Review Comments to the Author

Reviewer #1: This article provides an insightful and contextually appropriate qualitative study into healthcare providers’ experiences with 7.1% chlorohexidine (CHX) digluconate for umbilical cord care in Kiambu County, Kenya. Given Kenya’s public health goals for infant mortality and WHO guidelines for CHX in areas with high newborn sepsis risk, this topic is highly significant.

Methodologically, in-depth interviews and thematic analysis are appropriate, and the data collected clearly supports the core study findings. This study provides a useful insight into barriers (e.g., cost, availability, and knowledge gaps), facilitators (perceived efficacy and ease of use), and systemic concerns (training, policy compliance) that influence CHX adoption.

Meanwhile, this paper needs to be strengthened in these fields:

Strengthening Manuscript Structure:

Though the English of this paper is mostly clear, it needs somewhere to increase more clarity and conciseness. The results section should follow thematic frameworks and could be classified in several subheadings like barriers, facilitators, and knowledge gaps. Moreover, there is a lack of demarcation, like blending description, interpretation, and respondents’ quotations. There are some repeated sections in the discussion part, and it would be better if it could be synthesized with previous literature.

Lack of thematic analysis presentation:

Although NVivo was used for doing thematic analysis, presentation of the coding tree (arranging the data set in a theme/subtheme structure) is missing from this manuscript (only a part is shown in Table 1). Consider including a diagram to visually summarize key themes and their relationships to subthemes.

Using quotations in a standard way:

Quotes are useful for qualitative study, but they should not interrupt the flow of narrative. Some quotes are quite lengthy and should be condensed and narrowed down to better explain the key findings.

Lack of critical engagement in the discussion part:

The discussion part should be integrated with literature. The authors wrote the discussion part of this manuscript in a highly cited and descriptive manner. They often list other studies without deeply comparing study findings.

The abstract part is too detailed:

The abstract part of this article is too detailed, which could exceed PLOS ONE length expectations. It could focus on key findings and methods and brief recommendations.

Collusions should be sharply stated:

The conclusion part of this article is somehow reiterating findings. The article fails to provide a compelling conclusion or policy recommendation for future actions.

General recommendation: Major Revisions

This article may make a meaningful contribution to infant care interventions in low-resource settings once the above issues, notably the structure of results, depth of discussion, and clarity of language, are addressed.

Reviewer #2: In methodology better have a description on available formulations gel/ liquid form. SOP if any on how to use it, whether adequate training conducted by government or hospital authority on how to use and how frequently to apply CHX (once or multiple times}

HCW also faced interview through Kiswahili as well, it is better to mention whether questionnaire was validated in that language

Seems consent taken from hospital medical superintendent but here information lacking whether consent was taken from the individuals

Out of 2,600 HCWs only 38 were interviewed which may decrease the strength of the study

As it is a qualitative study it is better to mention about FGD, KII to get an overall idea to fulfill the objective of the study which seems missing here.

Existing cord care practice section

Please try to mention percentages in bracket after the term mentioned as majority/ many like that

Different facilities like 1 to 5 need to explain clearly

Please make it clear idea about recommendation about who to put CHX on umbilical cord; mother or health care provider?

Having high institutional delivery, please include the information about the responsible person to apply CHX while the baby is in hospital; health care provider/ mother where HCW can apply CHX considering safety and hygiene issue.

Reviewer #3: Thank you for submitting your manuscript. This manuscript explores an important topic, cord care practices and CHX implementation in a Kenyan county. While the study is timely and relevant, several critical areas require improvement to strengthen its scientific contribution. Detailed, section-wise suggestions are given below

Background:

• Add local epidemiological data: estimates of NNS, omphalitis, or mortality attributable to cord infections which would assist to understand the magnitude of the problem and strengthen the rationale for the study.

• This paragraph (third) is confusing. Contradictory evidence is described but not synthesized. what is the current consensus or lack thereof? Would recommend focus more the updated recommendation of WHO as these recommendations are made from multiple and multicountry studies and based on strong evidences.

• What was the specific evidence from Bangladesh study is missing here. Would suggest to clarify the conflict for clear understanding of the readers.

• "CHX protocol” is vague. Would recommend to specify if referring to 4% CHX gel, application frequency, or WHO guidance.

• The rationale is very weak. There is no mention of the systemic barriers (supply chain, training gaps, facility-level enforcement) that might present in the low-resouce settings which would provide context for CHX underutilization. Additionally, there is no mention whether any previous studies were conducted in Kenya or in the similar settings to understand HCW perspective, any gaps and limitation of those studies and what more this study may address particularly. Would strongly suggest to address these things to make the background scientifically sounder and more comprehensible.

Method:

• The design is identified as “exploratory qualitative,” but the justification for this choice is missing. Explain why this design was most suitable for assessing HCWs’ experiences.

• Lacks demographic details (cadre mix, gender, years of experience, urban vs rural distribution) which is crucial for assessing diversity and representativeness.

• Mention the qualification of JMG and BNO whether they received any qualitative interview training.

• Data management is briefly mentioned, but no info on data encryption, password protection, or cloud security for Kobo/NVivo files. would suggest to address.

Result:

• Add a table for clear understanding summarizing:

o Number of HCWs by cadre (nurses, pediatrician, pharmacist, MO, clinical officers, etc.)

o Facility type (Level 5/4/3, public/private/faith-based)

o Urban vs rural distribution.

Findings:

• Merge overlapping themes/sub-themes (e.g., “availability” and “access”) and synthesize common patterns across quotes rather than listing them sequentially.

• This findings section provides a rich, grounded understanding of cord care practices and CHX implementation challenges in a Kenyan county. However, the current presentation is verbose, and could benefit from clearer synthesis and critical reflection. Consider a thematic framework diagram to visually depict intensity and co-occurrence of themes across categories (e.g., advantages, barriers). With revisions focused on thematic clarity, and contextual framing, this section could substantially enhance the manuscript’s contribution to the research in newborn health.

• While quotes are tagged by anonymized ID, more information (e.g., facility type or cadre) would help interpret responses.

Discussion:

• Would suggest to emphasize what new insights this study offers beyond confirming previous finding. For instance, specific systems-level insights about devolution and supply chains in Kiambu.

• The discrepancy in reported CHX pricing between studies (e.g., Ksh. 40 vs. 200) requires deeper discussion. Is this due to procurement models, subsidies, or corruption?

• While lack of training is mentioned, the scope or standard of existing training programs is not elaborated. Are they nonexistent, outdated, or misaligned with WHO protocols?

Conclusion:

• The conclusion blends findings and recommendations without clearly differentiating them. This weakens the take-home message.

• "One healthcare facility actively practiced cord care against the National Guidelines” is serious but underexplaine. This needs clarification or should be omitted from the conclusion unless further elaborated.

• Consider including a final sentence with a stronger policy implication (e.g., need for CHX to be prioritized in county-level budgets, routine supervision, or national monitoring frameworks)

6. PLOS authors have the option to publish the peer review history of their article (what does this mean? ). If published, this will include your full peer review and any attached files.

**Do you want your identity to be public for this peer review?** For information about this choice, including consent withdrawal, please see our Privacy Policy .

Reviewer #1: **Yes: ** Md Taqbir Us Samad Talha

Reviewer #2: **Yes: ** Sanjoy kumer Dey

Reviewer #3: **Yes: ** Goutom Banik

---

## [Author Response · Author response to Decision Letter 1]

29 May 2025

Response to Reviewers

Dear Editor-in-Chief,

I hope this message finds you well. On behalf of my co-authors and myself, I would like to extend our sincere thanks for the opportunity to revise our manuscript, “...in Kiambu county, there are different things being done”: A Qualitative Study Exploring the Experiences of Healthcare Workers with 7.1% Chlorhexidine digluconate Cord Care in Kiambu County, Kenya (Manuscript Number: PONE-D-25-16839), following the comments by the reviewers of your esteemed journal of PLOS ONE.

We found the feedback provided to be incredibly insightful and valuable. The comments have guided us in refining our work and improving its clarity and rigor. We truly appreciate the time and effort that went into reviewing our submission. Please find below our rebuttal to each comment issued by the reviewers.

We look forward to continuing to engage with your office and we are confident that the manuscript will be better received owing to the constructive input we have received.

Thank you once again for your time and consideration.

Kind regards,

Corresponding author

Reviewers' comments:

Reviewer #1:

This article provides an insightful and contextually appropriate qualitative study into healthcare providers’ experiences with 7.1% chlorohexidine (CHX) digluconate for umbilical cord care in Kiambu County, Kenya. Given Kenya’s public health goals for infant mortality and WHO guidelines for CHX in areas with high newborn sepsis risk, this topic is highly significant.

Methodologically, in-depth interviews and thematic analysis are appropriate, and the data collected clearly supports the core study findings. This study provides a useful insight into barriers (e.g., cost, availability, and knowledge gaps), facilitators (perceived efficacy and ease of use), and systemic concerns (training, policy compliance) that influence CHX adoption.

Author’s response: Thank you for your thoughtful and encouraging feedback on our manuscript.

Meanwhile, this paper needs to be strengthened in these fields:

Strengthening Manuscript Structure:

Though the English of this paper is mostly clear, it needs somewhere to increase more clarity and conciseness. The results section should follow thematic frameworks and could be classified in several subheadings like barriers, facilitators, and knowledge gaps. Moreover, there is a lack of demarcation, like blending description, interpretation, and respondents’ quotations. There are some repeated sections in the discussion part, and it would be better if it could be synthesized with previous literature.

Author’s response: Thank you for your feedback. I have addressed the points you raised by improving clarity and conciseness throughout the paper. The results section has been reorganized to follow a clearer thematic framework with distinct subheadings such as barriers, facilitators, and knowledge gaps. Additionally, I have separated description, interpretation, and respondents’ quotations more clearly to avoid blending. Repetitive sections in the discussion have been synthesized with previous literature to enhance coherence.

Lack of thematic analysis presentation:

Although NVivo was used for doing thematic analysis, presentation of the coding tree (arranging the data set in a theme/subtheme structure) is missing from this manuscript (only a part is shown in Table 1). Consider including a diagram to visually summarize key themes and their relationships to subthemes.

Author’s response: Thank you for this comment. In lieu of the table 1, we have displayed the coding tree for this particular analysis, as below.

Figure 1: The clustered coding tree of transcripts from interview audio recordings. The numbers in brackets “References” column indicates the total number of individual references that mention that specific theme within the files.

Using quotations in a standard way:

Quotes are useful for qualitative study, but they should not interrupt the flow of narrative. Some quotes are quite lengthy and should be condensed and narrowed down to better explain the key findings.

Author’s response: Thank you for the helpful feedback. We agree with your suggestion and will revise the results section to improve narrative flow. Specifically, we will break up the findings into clearer sub-themes and shorten some of the longer quotes to ensure they support rather than interrupt the analysis. This has been done for the entirety of the results section.

Lack of critical engagement in the discussion part:

The discussion part should be integrated with literature. The authors wrote the discussion part of this manuscript in a highly cited and descriptive manner. They often list other studies without deeply comparing study findings.

Author’s response: Thank you for this observation. We have now reviewed and tailored our discussion to align with our recommendations based on the findings of our study. Kindly have a re-look at this section.

The abstract part is too detailed:

The abstract part of this article is too detailed, which could exceed PLOS ONE length expectations. It could focus on key findings and methods and brief recommendations.

Author’s response: I have shortened the abstract and it is within the word count of the journal.

Collusions should be sharply stated:

The conclusion part of this article is somehow reiterating findings. The article fails to provide a compelling conclusion or policy recommendation for future actions.

Author’s response: Thank you for this observation. We have now reviewed and tailored our conclusion to align with our policy recommendations based on the findings of our study. Kindly have a re-look at this section.

General recommendation: Major Revisions

This article may make a meaningful contribution to infant care interventions in low-resource settings once the above issues, notably the structure of results, depth of discussion, and clarity of language, are addressed.

Author’s response: Author’s response: Thank you for your thoughtful and encouraging feedback on our manuscript. We have attempted to address the issues within the manuscript and thank you for improving our work.

Reviewer #2:

In methodology better have a description on formulations gel/ liquid form. SOP if any on how to use it, whether adequate training conducted by government or hospital authority on how to use and how frequently to apply CHX (once or multiple times}

Author’s response: thank you we have included all this information in our methods section under the heading “Local context: CHX formulations, usage protocols and training”, as below.

“CHX for newborn umbilical cord care is available in two formulations: gel and solution [22]. Both forms are recommended by the WHO and are included in the Essential Medicines List as well as in postnatal care guidelines [22]. In line with this, Kenya’s Ministry of Health (MoH) endorses the availability and use of both CHX formulations [22]. Therefore, the national guidelines recommend 7.1% Chlorhexidine digluconate, which delivers 4% Chlorhexidine, for umbilical cord care both in health facilities and at home.

Immediately after delivery, a HCW applies CHX to the newborn’s umbilical cord using sterile gloves [22]. The antiseptic is applied to the base, stump, and tip of the cord, ensuring full coverage. For gel, the HCW uses an index finger to spread it; for solution, the dropper bottle is used to apply CHX directly. The CHX is not cleaned off after application.

In the postnatal period, CHX should be applied once daily until the seventh day or until the cord naturally detaches: whichever occurs first. Mothers and/or caregivers are instructed on how to apply CHX at home following the same procedures, but instead of sterile gloves, they are advised to thoroughly wash their hands before each application.”

HCW also faced interview through Kiswahili as well, it is better to mention whether questionnaire was validated in that language

Author’s response: The structured interview tool was developed and administered in English to guide the questions posed to healthcare workers (HCWs). Participants were allowed to verbally respond in either English or Kiswahili, depending on their preference.

Since the tool itself was only available in English, no formal validation or translation was done for Kiswahili; responses given in Kiswahili were translated as needed during transcription.

Seems consent taken from hospital medical superintendent but here information lacking whether consent was taken from the individuals

Author’s response: Yes, every participant consented to the study and we have detailed the same under our “Ethical Approval” section, as below

Prior to the initiation of data collection, participants were informed about the study's objectives and their role as respondents, including the importance of their continued involvement until the study's conclusion. This was conducted by JMG and BON. They were assured that participation was entirely voluntary and that they had the right to withdraw at any time. The researcher explained that all data and discussions would be recorded solely for the purposes of the study, and consent for this was obtained through the signing of a consent form.

Out of 2,600 HCWs only 38 were interviewed which may decrease the strength of the study

Author’s response: We appreciate the reviewer’s concern. The 2,600 healthcare workers represent the total number of HCWs across all cadres in the study setting. However, our qualitative interviews specifically targeted paediatricians, pharmacists, and nurses directly involved in cord care and postnatal care, which are the groups most relevant to our research question.

Unfortunately, specific numbers for HCWs working exclusively in cord care are not routinely collected or disaggregated in existing data systems. While the number of interviews may appear small in relation to the total HCW population, the participants were purposively sampled for their direct experience, which strengthens the relevance and depth of the findings.

We have also indicated that during the period of the study, there was a staff shortage in the county, which hindered data collection. Albeit we had an aim of 30 interviewees.

As it is a qualitative study it is better to mention about FGD, KII to get an overall idea to fulfill the objective of the study which seems missing here.

Author’s response:

Thank you for your comment. We have clarified in the study design section that the study employed an exploratory qualitative design using Key Informant Interviews (KIIs). A total of 38 HCWs were interviewed using semi-structured guides between 6th October and 9th November 2022 across six health facilities in Kiambu County. This is covered under the sub-heading of “Study Design”.

Existing cord care practice section

Please try to mention percentages in bracket after the term mentioned as majority/ many like that

Author’s response: Thank you for the suggestion. We have now included percentages in brackets following terms used.

Different facilities like 1 to 5 need to explain clearly

Author’s response: Thank you for your comment. I have clarified the descriptions of the different facility levels (levels 1 to 5) to ensure readers understand the distinctions between them in terms of services offered. This has been covered under “Study settings”, as below:

The county has 505 health facilities, comprising 108 public, 64 faith-based, and 333 privately owned facilities. In Kenya’s health system, Level 1 refers to community-based services (e.g., health promotion and preventive care), Level 2 to dispensaries, Level 3 to health centres, Level 4 to sub-county hospitals, and Level 5 to county referral hospitals. Maternity are typically offered at Level 3 (health centres) and above, including private and faith-based hospitals. Therefore, this study gathered insights from healthcare providers working in Level 3, 4, and 5 public facilities, as well as private maternity and faith-based hospitals. One of the Level 5 hospitals included in this study also functions as a teaching hospital for medical and nursing students.

Please make it clear idea about recommendation about who to put CHX on umbilical cord; mother or health care provider?

Author’s response: thank you we have included all this information in our methods section under the heading “Local context: CHX formulations, usage protocols and training”, as below.

“CHX for newborn umbilical cord care is available in two formulations: gel and solution [22]. Both forms are recommended by the WHO and are included in the Essential Medicines List as well as in postnatal care guidelines [22]. In line with this, Kenya’s Ministry of Health (MoH) endorses the availability and use of both CHX formulations [22]. Therefore, the national guidelines recommend 7.1% Chlorhexidine digluconate, which delivers 4% Chlorhexidine, for umbilical cord care both in health facilities and at home.

Immediately after delivery, a HCW applies CHX to the newborn’s umbilical cord using sterile gloves [22]. The antiseptic is applied to the base, stump, and tip of the cord, ensuring full coverage. For gel, the HCW uses an index finger to spread it; for solution, the dropper bottle is used to apply CHX directly. The CHX is not cleaned off after application.

In the postnatal period, CHX should be applied once daily until the seventh day or until the cord naturally detaches: whichever occurs first. Mothers and/or caregivers are instructed on how to apply CHX at home following the same procedures, but instead of sterile gloves, they are advised to thoroughly wash their hands before each application.”

Having high institutional delivery, please include the information about the responsible person to apply CHX while the baby is in hospital; health care provider/ mother where HCW can apply CHX considering safety and hygiene issue.

Author’s response: thank you we have included all this information in our methods section under the heading “Local context: CHX formulations, usage protocols and training”, as below.

“CHX for newborn umbilical cord care is available in two formulations: gel and solution [22]. Both forms are recommended by the WHO and are included in the Essential Medicines List as well as in postnatal care guidelines [22]. In line with this, Kenya’s Ministry of Health (MoH) endorses the availability and use of both CHX formulations [22]. Therefore, the national guidelines recommend 7.1% Chlorhexidine digluconate, which delivers 4% Chlorhexidine, for umbilical cord care both in health facilities and at home.

Immediately after delivery, a HCW applies CHX to the newborn’s umbilical cord using sterile gloves [22]. The antiseptic is applied to the base, stump, and tip of the cord, ensuring full coverage. For gel, the HCW uses an index finger to spread it; for solution, the dropper bottle is used to apply CHX directly. The CHX is not cleaned off after application.

In the postnatal period, CHX should be applied once daily until the seventh day or until the cord naturally detaches: whichever occurs first. Mothers and/or caregivers are instructed on how to apply CHX at home following the same procedures, but instead of sterile gloves, they are advised to thoroughly wash their hands before each application.”

Reviewer #3:

Thank you for submitting your manuscript. This manuscript explores an important topic, cord care practices and CHX implementation in a Kenyan county. While the study is timely and relevant, several critical areas require improvement to strengthen its scientific contribution. Detailed, section-wise suggestions are given below

Author’s response: Thank you for your thoughtful and encouraging feedback on our manuscript.

Background:

• Add local epidemiological data: estimates of NNS, omphalitis, or mortality attributable to cord infections which would assist to understand the magnitude of the problem and strengthen the rationale for the study.

• This paragraph (third) is confusing. Contradictory evidence is described but not synthesized. what is the current consensus or lack thereof? Would recommend focus more the updated recommendation of WHO as these recommendations ar

---

## [Editor Report · Decision Letter 1]

“...in Kiambu county, there are different things being done”: A qualitative exploration of healthcare workers’ experiences with cord care practices

PONE-D-25-16839R1

Dear Dr. Juttla,

We’re pleased to inform you that your manuscript has been judged scientifically suitable for publication and will be formally accepted for publication once it meets all outstanding technical requirements.

Kind regards,

Sojib Zaman, MBBS, MSc., PhD

Academic Editor

PLOS ONE
---

## [Editor Report · Acceptance letter]

PONE-D-25-16839R1

PLOS ONE

Dear Dr. Juttla,

I'm pleased to inform you that your manuscript has been deemed suitable for publication in PLOS ONE. Congratulations! Your manuscript is now being handed over to our production team.

Kind regards,

on behalf of

Dr. Sojib Zaman

Academic Editor

PLOS ONE